# Low-Cost Sensors Technologies for Monitoring Sustainability and Safety Issues in Mining Activities: Advances, Gaps, and Future Directions in the Digitalization for Smart Mining

**DOI:** 10.3390/s23156846

**Published:** 2023-08-01

**Authors:** Carlos Cacciuttolo, Valentina Guzmán, Patricio Catriñir, Edison Atencio, Seyedmilad Komarizadehasl, Jose Antonio Lozano-Galant

**Affiliations:** 1Civil Works and Geology Department, Catholic University of Temuco, Temuco 4780000, Chile; vguzman2015@alu.uct.cl (V.G.); pcatrinir2016@alu.uct.cl (P.C.); 2Department of Civil Engineering, Universidad de Castilla-La Mancha, Av. Camilo Jose Cela s/n, 13071 Ciudad Real, Spain; edison.atencio@pucv.cl (E.A.); joseantonio.lozano@uclm.es (J.A.L.-G.); 3School of Civil Engineering, Pontificia Universidad Católica de Valparaíso, Av. Brasil 2147, Valparaíso 2340000, Chile; 4Department of Civil and Environment Engineering, Universitat Politècnica de Catalunya, BarcelonaTech, C/Jordi Girona 1-3, 08034 Barcelona, Spain; milad.komary@upc.edu

**Keywords:** low-cost sensors, monitoring, safety, sustainability, mining, Raspberry Pi, Arduino, Internet of Things, digital mine, Industry 4.0

## Abstract

Nowadays, monitoring aspects related to sustainability and safety in mining activities worldwide are a priority, to mitigate socio-environmental impacts, promote efficient use of water, reduce carbon footprint, use renewable energies, reduce mine waste, and minimize the risks of accidents and fatalities. In this context, the implementation of sensor technologies is an attractive alternative for the mining industry in the current digitalization context. To have a digital mine, sensors are essential and form the basis of Industry 4.0, and to allow a more accelerated, reliable, and massive digital transformation, low-cost sensor technology solutions may help to achieve these goals. This article focuses on studying the state of the art of implementing low-cost sensor technologies to monitor sustainability and safety aspects in mining activities, through the review of scientific literature. The methodology applied in this article was carried out by means of the Preferred Reporting Items for Systematic Reviews and Meta-Analyses (PRISMA) guidelines and generating science mapping. For this, a methodological procedure of three steps was implemented: (i) Bibliometric analysis as a quantitative method, (ii) Systematic review of literature as a qualitative method, and (iii) Mixed review as a method to integrate the findings found in (i) and (ii). Finally, according to the results obtained, the main advances, gaps, and future directions in the implementation of low-cost sensor technologies for use in smart mining are exposed. Digital transformation aspects for data measurement with low-cost sensors by real-time monitoring, use of wireless network systems, artificial intelligence, machine learning, digital twins, and the Internet of Things, among other technologies of the Industry 4.0 era are discussed.

## 1. Introduction

### 1.1. Road to Smart Mining Considering Digitalization Technologies and Industry 4.0 Paradigm

Mining has developed and evolved over time, beginning as an artisanal activity and becoming industrialized at very sophisticated levels [1]. Undoubtedly, today, mining is a fundamental activity for the development of humanity, considering the high demand for commodities by the industries of digital electronic items and the components of devices to develop the industry of renewable energies [2]. Humanity has required a series of chemical elements to manufacture utensils, tools, artifacts, and machinery for its well-being and daily life. In this way, metallic (Cu, Ag, Au, Fe, C, among others) and non-metallic (Li, I, B, salts, among others) mining have been developed in different regions of the world, carrying out a series of deposit exploration and exploitation projects [3]. This is how copper (Cu) and lithium (Li), for example, are two key chemical elements worldwide to allow the development of the energy transition to renewable energies and to implement electromobility, respectively, contributing to sustainable development goals (SDGs) and thus achieve carbon neutrality by 2050 [4].

Mining projects have always been challenging, for example, being located in complex geographic locations, with extreme climates, rugged topographies, and the presence of surrounding communities and ecosystems [5]. Another challenge is its complex and multidisciplinary nature, due to the different areas that make up the systemic structure of a mining project, requiring the participation of professionals from various disciplines [6,7].

The mining industry, however, is quite traditional and somewhat slow to change technological systems because there is a scarce culture of assuming certain levels of risk and using new and innovative technological solutions that have not been widely tested [8]. A fully integrated automated system is a practical challenge for some mining companies, due to infrastructure limitations in communication, management, and data storage. Currently, there is a tendency for many mining companies to promote new technologies different from traditional methods, in order to rely on innovative techniques that accelerate progress towards a digital mine [9].

The first Industrial Revolution (Industry 1.0) began with the creation of the steam engine, which took place between 1760 and 1840. This made it possible to mechanize many processes and create distinct before-and-after eras. However, few people know that the first such machine was created for mining to pump water. The manufacture of machines generated the consumption of iron and coal, key minerals for its success. The Second Industrial Revolution (Industry 2.0), which took place between 1860 and 1920, is associated with electricity, online assemblies, and mass production. This set of factors facilitated, for example, the development of the automobile. One of the positive impacts of this Second Revolution was the strong boost in the consumption of commodities, especially iron, copper, oil, and nitrates, among others. Copper was fundamental for the massification of electricity for public use after 1870 and up to the present day. With electricity, the creation of new technologies seemed limitless. Between 1940 and 1970, the Third Industrial Revolution (Industry 3.0) materialized, driven mainly by computing, automation and robotics, lasers, and nuclear energy. Like the first revolutions, the Third Industrial Revolution brought with it a strong impact on the consumption of commodities, such as aluminum, oil, and uranium, and the last is essential for the development of nuclear energy [10,11].

It is in this context that Industry 4.0 signifies a paradigm of a new industrial revolution that unites advanced production and operations techniques with intelligent digital technologies to create a digital mine that would not only be interconnected and autonomous but could also communicate, analyze, and use data to drive more intelligent action back into the physical world [12,13]. It represents how smart and connected technology would be integrated into organizations, people, and assets, and is marked by the emergence of technological capabilities such as massive sensor deployment, data analysis, artificial intelligence (AI), machine learning (ML), Internet of Things (IoT), digital twins, cloud storage, 3D printing, virtual/augmented reality, 5G internet, automation, robotics, cybersecurity, nanotechnology, additive manufacturing, and advanced materials [14].

Addressing the challenges of future mining in the context of climate change, community claims for a pollution-free environment, and living in a complex and uncertain world requires the incorporation of information and communication technologies (ICTs) with advanced capabilities of (i) monitoring, (ii) control, (iii) optimization, and (iv) autonomous operation. These four capacities constitute an intelligent product, where each of these needs the previous one to be able to develop [15].

Security, storage, Internet of Things (IoT), big data, cloud computing, mobility, and robotics technologies are increasingly crucial to improve the productive factors of industries by impacting competitiveness, the added value of products and services, operational excellence, and export, but it requires a robust and specific base, which includes the sensorization systems or sensors [16].

In this context, real-time data collection technologies through sensors play a fundamental role in the development of Industry 4.0 applied to mining, requiring the development of sensors with efficiency, precision, speed, quality, and low cost in the data collection, and in this way, feed enough information to other intelligent systems within the digital mine [17]. By having the ability to analyze and interpret more information in real time, it will be possible to make proactive and intelligent decisions that benefit the production process, safety, and sustainability of the mining activity [18].

### 1.2. Use of Low-Cost Sensors for Monitoring Sustainability and Safety in Mining Activities

Currently, mining in its constant progress and evolution towards a responsible activity with its collaborators, neighboring communities, and the environment is making important efforts in its principles of sustainability and safety [19,20]. Mining is carried out in a territory, which has social and environmental dimensions that characterize it with an ancestral history that must be respected and incorporated into the projects. In addition, today, there are small, medium, and large mining companies worldwide that have thousands of collaborators who work both in corporate offices and in the field of mining operations and metallurgical plants [21].

Unfortunately, history tells us that mining has been marked by catastrophic events related to the lack of sustainability and lack of security. Events of ecosystem contamination, diseases in neighboring communities, and accidents of mining workers in underground and surface mining operations have allowed the mining activity to learn lessons and improve its productive system [22,23,24].

An important case to mention related to mining activity has been the shocking consequences of some failures of mine tailings storage facilities registered in the last decades, generating impacts and environmental damage, destruction of population centers, and deaths of human beings [25,26]. Some historical examples of these mine tailings storage facilities failures are: (i) Los Frailes, Spain, 1998, (ii) Baia Mare, Romania, 2000 (iii) Kolontar, Hungary, 2010, (iv) Mount Polley, Canada, 2014, (v) Fundao Samarco, Brazil, 2015, (vi) Brumadinho, Brazil, 2019, (vii) Jagersfontain, South Africa, 2022, and (viii) Williamson, Tanzania, 2022 [20,27].

On the other hand, we must mention how relevant the monitoring of safety aspects is to reducing the risks of accidents and fatalities in mining operations. Events of accidents in underground mines, surface mines, and metallurgical plants have promoted the application of management systems to improve safety and at the same time implement an adequate rescue strategy considering the best available technologies (BATs) and the best rescue practices (BRPs) [28,29,30]. Therefore, the accidents registered worldwide have had an impact on the safety standards applied in mining projects, in order to increase the levels of protection for workers, reduce risks, and eliminate fatalities [31,32].

It is in this context that the monitoring of sustainability and safety in mining has become a relevant value in the productive chain of the extractive activity of metals, being one of the fundamental pillars of corporate social responsibility (CSR) [33]. To assess and measure the performance of mining companies in sustainability and safety aspects, a series of indicators are being implemented, such as the Environmental, Social, and Governance (ESG) indicators for sustainability and Key Performance Indicators (KPIs) for safety [34,35].

The performance of sensor systems is strongly influenced by the environmental conditions in which the device is immersed. This is how temperature, air humidity, water quality, oxidation/corrosion conditions, soil chemical composition, and geographical altitude, among others, are variables that must be considered in mining projects that are generally in complex geographies with extreme climates [36,37].

For this reason, the implementation of sensors requires a periodic inspection in the field by trained personnel to evaluate their performance and at the same time carry out preventive maintenance in case their operation is altered by environmental conditions [38]. In this way, sensor changes, sensor cleaning, or implementation of protection systems are carried out periodically. Although the aim is to digitize mining operations and have measurements in real time, it is still necessary to inspect the sensor systems to verify in situ that the measurements are being carried out properly without alterations [39,40].

Technology is increasingly being perfected with more compact and low-cost sensor devices that replace current conventional equipment in different areas, low-cost sensors present a viable alternative as a substitute for current technologies in monitoring different variables in order to get data [41,42]. In the current framework of their use, these devices are available on the market and with a wide variety of operating principles, so the standardization and calibration of their application are still under development [18].

Considering the above, it is that low-cost sensor solutions have emerged, in order to improve efficiency in data collection, increase the quality of measurements, improve the speed of information collection, and reduce implementation costs [43,44]. For this reason, the electronic industry with different devices has provided alternatives for the sensorization of industrial processes, such as Arduino, and Raspberry Pi, among others. It is essential to mention that Raspberry Pi is a single-board computer, based on a microprocessor. In the same sense, Arduino is an electronics prototyping platform based on a microcontroller. The recent publications found in the literature review of low-cost sensors used in mining reveal a growing interest among researchers in utilizing devices such as Arduino and Raspberry Pi. These publications demonstrate the increasing attention and adoption of these devices in the mining, renewable energy, and agricultural fields. González et al. [45] present a monitoring system for visualizing the operation of a Lithium-ion Battery (LiB) using Internet of Things (IoT) technology. The system utilizes Grafana software hosted on a Raspberry Pi to provide real-time graphical and numerical information about various LiB magnitudes. Martin et al. [46] propose a novel system that utilizes artificial vision and a wireless sensor network based on IEEE 802.15.4 technology, with a Raspberry Pi as a key component, to track the maximum power point in solar photovoltaic systems under partial shading conditions. Experimental tests validate the system’s effectiveness, achieving a maximum power point tracking efficiency higher than 99%, even in partial shading conditions. Sangjan et al. [47] demonstrated the development of a Raspberry Pi-based sensor system for automated in-field monitoring to support crop breeding programs. The system integrates a microclimate sensor and multiple cameras (RGB and multispectral) to capture crop image data at user-defined time points throughout the season. Singh et al. [48] developed a novel system based on a low-cost chip-level colorimeter for detecting nitrogen and phosphorus concentrations in soil, aiming to restore optimal soil fertility, with an Arduino as the main control unit [45,46,47,48]. The low cost, portability, and ease of use make it possible for the people who work in the mining operation to participate in the monitoring of variables of interest within their daily activities [49].

Therefore, it is possible to mention that low-cost sensors have been proposed as a complement to existing conventional monitoring networks, to increase spatiotemporal resolution and provide real-time information on the level of exposure of production systems, environment of labor, and environment of mining operations [8,9]. The great challenge in the use of these low-cost sensors is to generate sufficiently reliable data to be considered in decision making for the management of the operations of an intelligent digital mine [5].

This review article has the following content structure for the proper understanding of the research: (i) Introduction: Context and scope of the research, (ii) Materials and Methods: Resources/materials considered in the research and methodological procedure applied in the research, (iii) Results: Presentation of graphs and tables with results, (iv) Discussion: Findings obtained in the research and (v) Conclusions: Lessons learned and recommendations.

### 1.3. Aim of the Article

This article focuses on studying the state of the art of the implementation of low-cost sensor technologies to monitor sustainability and safety aspects in mining activities, through the review of the scientific literature available on Web of Science (WoS) and Scopus. The methodology applied in this article was carried out using the Preferred Reporting Items for Systematic Reviews and Meta-Analyses (PRISMA) guidelines and the generation of scientific maps using the visualization software VOSviewer. For this, a 3-step methodological procedure was implemented: (i) Bibliometric analysis as a quantitative method, (ii) Systematic review of literature as a qualitative method, and (iii) Mixed review as a method to integrate the findings found. In this review of scientific literature, the following research questions (RQ) are considered:RQ1: How is knowledge about low-cost sensors applied to mining clustered, and how have they evolved over time?RQ2: What microcontroller technologies are implemented in low-cost sensors applied in mining?RQ3: What are the uses of low-cost sensors applied in mining?RQ4: How is the operation of low-cost sensors, manual or automatic for monitoring mining activities?RQ5: What kind of low-cost sensor technology is applied for monitoring mining activities?RQ6: How is the connectivity of low-cost sensors for monitoring mining activities?RQ7: What are the areas within the mining operation where low-cost sensors are applied to monitor mining activities?RQ8: What are the main engineering specialties where low-cost sensors are applied for mining monitoring?RQ9: What are the main application objectives of low-cost sensors implemented in mining?RQ10: What are the technologies related to Industry 4.0 that are currently being applied in low-cost sensors for monitoring mining activities?

Finally, according to the results obtained, the advances, gaps, and future directions in the implementation of low-cost sensor technologies for use in a digital mining environment are exposed, considering real-time monitoring, the use of artificial intelligence (AI), machine learning (ML), Internet of Things (IoT), digital twins, among other technologies linked to Industry 4.0.

## 2. Materials and Methods of Literature Review

### 2.1. Materials

#### 2.1.1. Web of Science and Scopus Scientific Databases

To develop this research, the scientific databases of Web of Science (WoS) and Scopus have been selected as the first source of information, as they are one of the most complete and relevant databases worldwide, which have a series of journals and data sources related to sensors and information digitization technologies. For the analysis of this publication, articles and review-type scientific publications have been considered, all of which were published in English from 1981 to 2023.

#### 2.1.2. MS Excel and VOSviewer Software

To develop the systematization, analysis, and processing of the information in the present investigation, the software MS Excel and VOSviewer have been used. The first is to systematize, analyze, and process the information through tables and graphs, while the second is to process the information through scientific maps.

### 2.2. Methods

#### 2.2.1. Bibliometric Analysis and Systematic Content Review

The methodology applied in this article was carried out using the Preferred Reporting Items for Systematic Reviews and Meta-Analyses (PRISMA) guidelines and the generation of scientific maps using the visualization software VOSviewer.

The PRISMA reporting guideline is the result of a consensus of experts in systematic reviews [50]. The PRISMA statement is a set of evidence-based items that systematic reviewers can use to report systematic reviews and meta-analyses. The PRISMA guideline provides recommendations that help address writing systematic reviews by helping systematic reviewers to report the rationale transparently, fully, and accurately for conducting the review [50].

According to the PRISMA methodology, a flowchart must be provided to represent the elimination steps, which are: “identification, detection, eligibility, and final inclusion” [50]. The study removal process is graphically represented in Figure 1.

For this, a 3-step methodological procedure was implemented as shown in Figure 2: (i) Bibliometric analysis as a quantitative method, (ii) Systematic review of literature as a qualitative method, and (iii) Mixed review as a method to integrate the findings found [51,52,53]. In this study, the 10 research questions mentioned in the introductory chapter of this article are considered.

In Figure 2, it is possible to appreciate the stages, research tools, activities, and deliverables developed in the methodology applied in this research. It is observed that the three major stages of the methodological procedure applied in this research are: (i) interrelated sensor monitoring study in the mining domain, through data acquisition, mapping, and bibliometric study, to later obtain the database of articles and maps of co-occurrence of keywords and cluster analysis, (ii) identification of low-cost sensors for monitoring in mining, systematic review, identification of monitoring domains of low-cost sensors, and then obtain listing of identified subdomains, and (iii) answers to the research questions, through an analysis of the integrated systematic review of bibliometrics, to finally obtain answers to the research questions and find the most relevant scopes of application of low-cost sensors [51].

Co-occurrence is the occurrence of two or more keywords that are very close to each other. Co-occurrence maps are often used as a classification signal, since they can help to understand the relationship between different terms and concepts, grouping them into groups or clusters and showing the words in different sizes and colors [54]. Therefore, co-occurrence maps focus on the visualization of distance-based bibliometric networks that support a large amount of metadata. The importance of a co-occurrence map is to identify the conceptual and thematic structure of a certain scientific domain [55].

#### 2.2.2. Article Selection Process

To develop the selection process of scientific articles to be analyzed in this research, a search was carried out based on a set of keywords, using Boolean operators, applied both to the Web of Science (WoS) and Scopus [56]. Therefore, it was necessary to define a series of key concepts within the theme to be developed in this research. The keywords selected for this study were: (i) sensor, (ii) low-cost sensor, (iii) monitoring, (iv) mining, (v) Arduino, and (vi) Raspberry Pi. Once the keywords were chosen, it was possible to proceed to make four combinations of keywords using the Boolean AND operator as shown below in Table 1:

It is essential to mention that the primary focus of this work is specifically on low-cost sensors. Given that the inclusion of the keyword “digitization” would have significantly expanded the scope of the search and potentially yielded a larger quantity of scientific articles, a deliberate decision was made to maintain a narrow focus on low-cost sensors.

Once the results were obtained through the search strategies, the scientific articles that applied the search criteria were grouped and selected. It is worth mentioning that two search strategies were applied, starting from the most general (sensor, monitoring, and mining) to the most specific (low-cost sensor, monitoring, and mining).

Finally, with the scientific articles selected, the data extraction form was evaluated, obtaining information from the metadata analysis perspective (DEM) and the content analysis perspective (DEC) as shown in Table 2.

## 3. Results

The results obtained in this investigation are presented below:

### 3.1. Article Screening Process

To develop the selection process of scientific articles to be analyzed in this research, an analysis was carried out in stages considering exclusion criteria (EC) to select scientific articles [57]. In Figure 3, it is possible to observe the procedure applied together with its results.

Figure 3 represents the extraction of documents from the Web of Science (WoS) and Scopus databases considering the combinations of keywords mentioned above, with which 5254 articles were obtained. A first filter or exclusion criterion 1 (EC1) was applied to these documents in order to eliminate duplicate articles, leaving a quantity of 4218 articles. Then a second filter or exclusion criterion 2 (EC2) was applied, which consisted of reading the abstract of the articles to determine which ones are related to sensors applied in mining, giving a result of 1066 articles. Finally, a last filter or exclusion criterion 3 (EC3) was carried out, which consisted of searching the 1066 articles that contained the keywords “Arduino, Raspberry Pi, and Low-cost sensor”, finally leaving 51 articles. These 51 selected articles deal specifically with “low-cost” sensors applied in mining, which will be subject to a more detailed analysis in the following pages of this research.

### 3.2. Bibliometric Analysis Results for Sensors Monitoring Mining (1066 Articles Selected)

#### 3.2.1. Annual Quantitative Distribution of Literature

Next, in Figure 4, the results obtained for the 1066 articles obtained as part of the selection process mentioned above are presented.

Figure 4 indicates the number of articles related to mining sensors published per year, considering the sample of 1066 articles, from which this graph shows that the year with the most publications was 2022, while the least published was 1982. Despite some variations, it can be considered that over the years, the number of publications and interest in the subject have increased as well. It should be considered that the sample was taken in February 2023; therefore, said year does not include all the publications generated.

#### 3.2.2. Country Distribution of Selected Articles

Next, Figure 5 presents the frequency of contributions by country for the 1066 articles selected in the first iteration.

Figure 5 represents the sample of the 16 countries with the largest number of publications on the subject. The number of documents produced by the country is mainly concentrated in China, the United States, India, Poland, Canada, Australia, Germany, Russia, and Brazil, which represent approximately 80% of the 1066 articles, while Italy, Chile, South Africa, Spain, the Czech Republic, Japan, and South Korea also stand out in percentage of publications.

#### 3.2.3. Quantitative Analysis of Number of Citations

Below, Figure 6 presents the results obtained for the productivity of citations of scientific publications by country in recent years, taking into account the 1066 selected articles.

The countries with the largest number of documents cited are shown in Figure 6, where approximately 80% of the articles belong to China, Canada, India, the United States, Australia, Hong Kong, Poland, Japan, and Germany. In addition, China stands out as the country with the largest number of articles cited worldwide. Other countries such as Italy, Chile, France, South Africa, South Korea, and the United Kingdom complete the list of the 16 most cited countries, which account for approximately 93% of the total citations.

#### 3.2.4. Quantitative Analysis of Document Type

Next, in Figure 7, the results obtained for the distribution of document types of scientific publications in recent years are presented, considering the 1066 selected articles.

The pie chart in Figure 7 shows the types of documents found in the databases, of which scientific articles predominate with 540, followed by conference papers with 466, to complete all the types of documents found. A total of 33 review-type papers, 15 book chapters, and 12 documents are in the other category. This means that the knowledge of sensors applied to monitoring in mining is being published both at a scientific level and at the level of advances of the state of the practices of the mining industry in conferences.

### 3.3. Systematic Content Review Results for Sensors Monitoring Mining (1066 Articles Selected)

The bibliometric metadata obtained as a result of the search for the 1066 selected articles were processed in VOSviewer to produce a co-occurrence map without considering the time scale. The minimum occurrence value was two keywords, which means that a keyword appears on the map when two articles cite it. The resulting map shows five clusters (represented in green, purple, blue, red, and yellow), as shown in Figure 8.

In the co-occurrence map in Figure 8, five clusters were generated, which are represented by different colors. Within the keywords, the ones that stand out the most are monitoring, coal mines, and sensors. Table 3 presents an interpretative summary in Figure 8, where it is possible to appreciate the theme applied by cluster and prominent keywords of the co-occurrence analysis.

Considering the cluster interpretation of Table 3, it is possible to find as a finding that the development of the application of sensors has a strong application trend in underground mining for structural monitoring, focused on the carbon commodity, to develop later advances in sustainability monitoring and safety of mining activities.

On the other hand, the bibliometric metadata resulting from the search of the 1066 selected articles were processed in VOSviewer to produce a co-occurrence map considering the time scale. The resulting map shows five clusters (represented in green, blue, and yellow) in terms of the year in which the selected articles have been published, as shown in Figure 9.

In this co-occurrence map in Figure 9, it is possible to appreciate the time scale in which these concepts were generated, the most recent being the Internet of Things (IoT), digital storage, and unmanned aerial vehicles (UAV). While the oldest are sensors, coal mines, and mining, among others. Finally, Table 4 presents an interpretative summary in Figure 9, where it is possible to appreciate the theme that applies by cluster and prominent keywords of the co-occurrence analysis.

Considering the cluster interpretation of Table 4, it is possible to find as a finding that the evolution over time of the application of sensors has a tendency to start with issues of networks or sensor arrays, then evolve and develop means of high wireless connectivity, and finally develop technologies towards the total digitization of information.

### 3.4. Bibliometric Analysis Results for Low-Cost Sensors Monitoring Mining (51 Articles Selected)

#### 3.4.1. Annual Quantitative Distribution of Literature

Below, Figure 10 presents the results obtained for the number of scientific publications in recent years, taking into account the 51 selected articles.

Figure 10 represents the number of publications per year considering the sample of 51 documents dealing with low-cost sensors, where the first publication dates were in the year 2004 and the peak of publications was the year 2020. It should be noted that the sample was taken in February 2023, so exact data on the number of publications generated during the course cannot be obtained of said year. The year 2021 shows a drop in the number of publications, probably referring to the global pandemic of COVID-19 [58,59].

#### 3.4.2. Country Distribution of Selected Articles

Below, Figure 11 presents the results obtained for the productivity of scientific publications by country in recent years, considering the 51 selected articles.

The graph in Figure 11 shows the number of documents on low-cost sensors applied in mining published by country, where the countries with the highest number of publications are China and India. This graph shows all the countries that produced the 51 documents on low-cost sensors applied in mining. Within the accumulated 80% of the publications generated, we find the following countries: China, India, Australia, United States, South Korea, Canada, Italy, Spain, and Brazil [60,61].

#### 3.4.3. Quantitative Analysis of Number of Citations

Next, in Figure 12, the results obtained for the productivity of citations of scientific publications by country in the recent years are presented, considering the 51 selected articles.

The graph in Figure 12 shows the number of citations by country, where China stands out with approximately 35% of the total citations and Australia with 23% of the citations. The countries that complete 80% of the accumulated appointments are India, Canada, Poland, and South Korea [62,63].

#### 3.4.4. Quantitative Analysis of Document Type

Next, in Figure 13, the results obtained for the distribution of document types of scientific publications in the recent years are presented, considering the 51 selected articles.

The pie chart in Figure 13 identifies the types of documents in the 51 low-cost sensor publications, leading the article with 29 documents, followed by conference papers with 18, and further back, we find 3 review-type papers and 1 chapter of the book. This indicates that advances in low-cost sensor applications applied in mining are being published in scientific journals [64,65].

### 3.5. Systematic Content Review Results for Low-Cost Sensors Monitoring Mining (51 Articles Selected)

#### 3.5.1. Keyword Co-Occurrence Analysis

The bibliometric metadata obtained as a result of the low-cost sensor monitoring mining search was processed in VOSviewer to produce a co-occurrence map without considering the time scale. The minimum appearance value was two keywords, which means that a keyword appears on the map when two articles cite it. The resulting map shows five clusters (represented in green, purple, blue, red, and yellow), as shown in Figure 14.

When making the co-occurrence map with the keywords of the 51 documents shown in Figure 14, which deal with low-cost sensors applied in mining, four clusters were generated, which are differentiated by the colors yellow, blue, red, and green. The keywords that stand out are monitoring, Internet of Things, coal mines, costs, and air quality [66,67]. Table 5 shows the clusters and keywords that apply to the interpretation of the respective clusters.

Considering the cluster interpretation of Table 5, it is possible to find as a finding that low-cost sensors have a trend of application in mining to monitor environmental, safety, and occupational health aspects. In addition, the highlighted is the use of technologies linked to Industry 4.0.

The bibliometric metadata resulting from the low-cost sensor monitoring mining search considering the 51 selected articles were processed in VOSviewer to produce a co-occurrence map considering the time scale. The minimum occurrence value was two keywords, which means that a keyword appears on the map when two articles cite it over time. The resulting map shows five clusters (represented in green, blue, and yellow), as shown in Figure 15.

The graph in Figure 15 shows the time scale of the keywords with the highest co-occurrence of the 51 documents obtained after applying the last filter or exclusion criterion 3 (EC3). Within the oldest, it is possible to find coal mines, gas detectors, and coal mine safety, while in the most recent, keywords low-cost sensors and air quality stand out [68,69]. Table 6 shows the cluster and keywords that apply to the interpretation of the respective cluster.

Considering the cluster interpretation in Table 6, it is possible to find as a finding that the evolution over time of the application of low-cost sensors has a tendency to start with environmental issues, then evolve towards safety, and finally develop Industry 4.0 technologies.

#### 3.5.2. Content-Based Data Perspective Analysis

Next, in this chapter of this review article, the answers to the research questions raised through an exhaustive content analysis considering the 51 selected articles are presented.

The data on the use of different microcontroller and/or microprocessor boards are represented in the graph in Figure 16, where 24 documents that are equivalent to more than 50% implement microprocessor technologies, mainly highlighting Arduino with 13 and Raspberry Pi with 5, 22 documents do not specify any type of microprocessor technology in monitoring systems used in mining [70,71].

It is essential to mention that Raspberry Pi is a single-board computer, based on a microprocessor. In the same sense, Arduino is an electronics prototyping platform based on a microcontroller.

Figure 17 shows the main applications of sensors in mining monitoring, highlighting air quality monitoring by a wide margin with 26 documents, followed by slope monitoring and health monitoring with 3 documents each of mining workers. It is appreciated that the focus has been on air quality monitoring [72].

Figure 18 represents the form of data collection from the sensors, with automatic data collection considered in real time and manual data collection not in real time. The graph shows that in 44 documents, the data are automatically collected in real time, which is the predominant way in the documents reviewed; only in two documents, the data collection from the sensors is performed manually, not in real time [73].

The graph in Figure 19 represents the information on the different types of sensors used in the analyzed literature: gas sensors lead the count in 15 mentions, temperature sensors continue with 14 mentions, and then humidity sensors continue with 8 mentions. There is a tendency to monitor aspects related to air quality in mining work environments [74].

The type of data transmission implemented in the connectivity of the sensors is shown in Figure 20, where the three main types of data transmission are Zigbee with 15 applications, Bluetooth with 12 applications, and WiFi with 10 applications, mainly the form of wireless connection [75].

In the item of the area of application of low-cost sensors shown in Figure 21, the use in underground mines stands out in 25 documents, followed by open-pit mines in 11 documents, and then sensors used in all mining areas in 7 documents, mineral transport with 3 documents, processing plant with 2 documents, and ending with 1 document the mine tailings storage facility area [76].

The engineering specialties where the low-cost sensors are applied shown in Figure 22, these are led by the environmental area in a vast majority with 32 mentions, followed by the structural area with 14 mentions, while the least mentioned areas are electronics and geophysics with only one mention each [77,78].

The circular graph in Figure 23 represents the main objectives of the application of low-cost sensors, which are grouped into 3 categories, mainly highlighting safety with 71 mentions, sustainability with 29 mentions, and structural integrity with 9 mentions, and it should be noted that some documents mentioned more than one objective in the application of sensors [79,80].

From the documents analyzed, the review of the implementation of technologies related to Industry 4.0 applied to low-cost sensors was carried out. Figure 24 clearly shows that the Internet of Things is the most applied technology in low-cost sensors with 15 mentions, followed by machine learning (ML) with 5 mentions. However, 22 of the documents reviewed do not specify Industry 4.0 technologies. Some documents implemented more than one type of technology related to Industry 4.0 [81,82].

Figure 25 shows some examples of the use of sensors in mining under an Industry 4.0 paradigm to monitor key variables for sustainability, productivity, and safety performance in different activities carried out in mining operations.

The implementation of the use of these different types of sensors in mining activities will require complementing it with other Industry 4.0 technologies, such as the Internet of Things (IoT), machine learning (ML), and artificial intelligence (AI). In addition, another important aspect to consider is providing adequate connectivity capacity, either with short-range systems such as Bluetooth and Wi-Fi or long-range systems such as Zigbee, XBee, LoRa, and LTE, among others.

### 3.6. Comparision of the Main Characteristics of the Articles Selected Dealing with Low-Cost Sensors Monitoring Mining

A comparison of the main characteristics of the articles selected is summarized in Table 7. This table includes: (1) Type of microprocessor, (2) Type of operation, (3) Data connectivity, (4) Mining areas (5) Engineering specialty, (6) Application objective, (7) Industry 4.0 technology, and (8) Cost in USD.

The analysis in Table 7 reveals the following results:Regarding the type of microprocessor, 48% of the cases are not specified, while 28% correspond to Arduino, 11% correspond to Raspberry Pi, and the remaining 13% correspond to other types of microprocessors. It is possible to notice that there is still no clear tendency to define the type of microprocessor for low-cost sensors applied in mining.When studying the type of operation of low-cost sensors, 96% corresponds to automatic applications with data collection in real time, and the remaining 4% are manual data collection applications not in real time. This teaches us that the current trend is to monitor mining activities in real time.Regarding the connectivity of low-cost sensors, 25% corresponds to Zigbee, 20% corresponds to Bluetooth, 17% is WiFi, and 38% corresponds to other applications. This shows the trend of using low-cost sensors wirelessly in mining monitoring applications.When analyzing the area of the mining project where the low-cost sensors are applied, 51% are applied in an underground mine, 22% in an open pit mine, 14% in all areas of a mining project, 6% in mineral transport, 4% in metallurgical plants, and 1% in mine tailings storage facilities. This tells us that currently the emphasis of monitoring in mining with low-cost sensors is focused on underground mine operations.When studying engineering specialties, the discipline with the greatest application is environmental at 41%, then structural at 18%, followed later by geotechnics at 14%, then civil at 10%, then mining at 9%, and others at 9%. This shows us that sustainability aspects are a priority in mining considering the use of low-cost sensors.Considering the monitoring objectives in low-cost sensors, 65% of the cases correspond to safety, while 27% correspond to sustainability and 8% correspond to structural integrity. This tells us that mining safety monitoring is essential in order to ensure the life and health of its collaborators.Regarding the technologies linked to Industry 4.0, 42% of the cases are not specified, On the other hand, 28% of the cases correspond to the Internet of Things (IoT), 9% to machine learning (ML), and 21% to % to others. This shows that low-cost sensors are linked to the Internet of Things but shows that integration with other Industry 4.0 technologies is still lacking.The analysis of the cost reveals that, although all applications claim to have a reduced price, this characteristic is rarely documented in the literature [99,100].

## 4. Discussion

### 4.1. Advances

According to the results obtained in this research, an upward trend is observed in the number of scientific publications related to the subject of monitoring with sensors in mining, producing a slight drop in the years 2020 and 2021, is expected, due to the global COVID-19 pandemic [101,102].

The Industry 4.0 paradigm compared to the previous transitions 1.0, 2.0, and 3.0 has generated more dizzying and disruptive changes in mining, and currently, the world is in a stage of full development. Although in previous transitions, productivity was the aspect that had a substantial improvement, currently in the Industry 4.0 paradigm, sustainability and safety are the key principles of the mining business [10,11].

The most productive and leading-edge countries in terms of scientific publications related to low-cost sensor applications are in order of productivity from highest to lowest: China, India, Australia, the United States, and South Korea. These countries also agree regarding the number of citations to scientific publications related to low-cost sensor applications.

Considering the results obtained in this research, both China and India are the countries that have the largest number of scientific articles published on low-cost sensors applied in the mining industry. In the case of China, there is a well-diversified mining industry with metallic and non-metallic mining of different commodities, while India, although it does not have a strong mining industry, its researchers are focused on developing publications on the subject. Another country that is also a protagonist, behind China and India, in publications on low-cost sensors applied to mining is Australia, a mining country with various metallic and non-metallic mining operations. The large number of publications and citations for this country is striking. This demonstrates the interest of these countries, both governments and private companies, in investment plans for economic resources to finance research, development, and innovation in sensor technologies.

On the other hand, considering the results obtained in this research on low-cost sensors applied in mining, the scientific publications that stand out the most with the greatest number are article-type documents, followed by conference papers, then review-type documents, and finally, book chapters. This shows that novel and unpublished research is being developed in which the results and findings found are being presented through article-type documents.

It is possible to mention that both the technological elements of Arduino microprocessors and Raspberry Pi devices are pointed out in publications related to low-cost sensors applied in mining, but there is also a significant number of publications that do not yet apply this type of technology in their mining sensors systems.

By far, the most popular application of low-cost sensors applied to mining is related to sensors to monitor air quality, whether in underground or open-pit mines. A large number of publications mention or cite them, and to date, there has been great progress and development in this application.

On the other hand, it is possible to mention that, in most cases of the application of low-cost sensors in mining, the measurements or data collection is through real-time or automatic measurements, leaving manual measurements behind automatic measurements.

The most used types of low-cost sensors according to this bibliographic review correspond to gas detection sensors and temperature measurement sensors in mining operations.

Regarding the connectivity and form of data transmission, as a result of this research, the uses of Zigbee, Bluetooth, and WiFi demonstrate the massive use without the use of cables but rather of wireless data transmission.

Another finding found in this investigation is that the area of the mine or mining deposit where work is carried out, whether in an underground mine and/or surface mining in the open pit, is the most developed in terms of implementation of low-cost sensors, and underground mining is the more advanced. Other areas such as the metallurgical process plant, transportation, and mining waste deposits show more delays in the implementation of low-cost sensors.

Considering the engineering specialties involved in mining operations, the specialties with the greatest application of low-cost sensors in mining are environment, structures, and geotechnics.

Considering, on the other hand, the objectives of the application of low-cost sensors in mining, the safety objective stands out leading, followed by sustainability, and lastly, structural integrity. This demonstrates the importance that the mining industry is giving to these aspects in the daily tasks of its operations.

In addition, it is possible to mention according to the results obtained that there is a massive use of remote sensors through unmanned autonomous vehicles (UAV) such as drones and/or satellites, which allow for providing topographic information or aerial photographs as satellite images with high resolution.

Finally, considering the information and communication technologies (ICTs) considered in Industry 4.0 linked to low-cost sensors applied in mining, the following stand out: Internet of Things (IoT), machine learning (ML), and automation.

### 4.2. Gaps

There is a gap in the publication of advances in Industry 4.0 issues in the field of mining, since there is not enough information published in scientific databases. It is known that information and communication technologies (ICTs) are part of the Industry 4.0 paradigm if they are being applied by mining companies worldwide, but mainly part of said knowledge and experiences, and lessons learned is published or shared in technical conferences, not being published in a massive way in journals and scientific magazines. It is also known that many world-class mining companies have research/development areas where innovation in mining-metallurgical processes and labor quality of their collaborators are encouraged, where many of the experiences developed are not shared, leaving the know-how only within these companies and their direct collaborators.

This is how some of the new and largest copper mining projects worldwide are found in Chile and Peru, which are advancing rapidly towards the digital transformation of their operations, to become digital mines and are, today, at the forefront of the Industry 4.0 era. The Chilean copper mining projects under this premise are: Quebrada Blanca Phase II (Teck), Spence (BHP Billiton), Escondida (BHP Billiton), Gabriela Mistral (Codelco), El Teniente (Codelco) and underground mine of Chuquicamata (Codelco). In the case of Peru, it is possible to mention the mining project Quellaveco (AngloAmerican), Antamina (BHP Billiton, Glencore, Teck), Cerro Verde (Freeport McMoran), and Yanacocha (Newmont) [103,104,105,106].

Although in the mining industry, not all operations are large-scale, it is also important to consider small and medium-sized mining. Although the economic resources to invest in innovation are more limited in small and medium-sized mining, low-cost sensor solutions are attractive to be implemented. In many cases, prototypes of new sensor technologies are tested and validated in small and medium-sized mining operations, these mining operations being places for generating knowledge, learning, and relevant practical experience [107].

In this context, another gap is visualized, which is the lack of proactive linkage between academia and industry to carry out joint research and thus subsequently publish the results in scientific journals or magazines, not only develop papers, but also go on the path towards innovation, and, for example, develop research patents for products, services or prototypes created.

On the other hand, an additional relevant gap that is visualized according to the results obtained in this research is how fast and dizzying the current process of digital transformation towards Industry 4.0 in mining, which will produce many changes in jobs in activities mining companies, eliminating some current functions and generating new jobs with a focus on new technologies [108]. At this manner, it is seen that older workers will have difficulties adequately understanding and managing some new technologies, this being an advantage for younger professionals such as data scientists, telematic engineers, and programmers, among others, raised and educated in a digital age.

According to the results found in this research, there is information on the use of low-cost sensors in coal mining, with various applications, but mainly related to safety monitoring in underground mines. In this context, there is a significant gap in scientific literature due to the scarcity of scientific publications on the application of technologies in the mining of other commodities such as lithium, copper, gold, silver, iron, uranium, and rare earth elements (REEs), among others.

On the other hand, considering the accidents of mine tailings storage facility dam failures and mine tailings spills, according to this investigation, there are not enough scientific articles to date that address this problem to promote a more responsible and controlled management of mine tailings through the massive use of low-cost sensors.

In the specific case of mining, considering the characteristics of complex and multidisciplinary projects, another gap identified is the lack of comprehensive and holistic work, for which the focus should be on improving information interoperability capabilities. Interoperability is defined as the ability of information and communications technology systems, and the business processes they enable to exchange data and allow information and knowledge to be shared, which in turn makes it possible for organizations and systems to work together without problems, that is, to inter-operate.

### 4.3. Future Directions

Undoubtedly, sensors are the basis that supports all the development of digitization in the mining industry, and over time, it will be a more massive and diversified activity, which will allow mining activity to evolve, making it more resilient and versatile to the changes that experience society in a complex and uncertain world, for example, due to the effects of climate change.

An important aspect to consider for the success of the application of sensors in mining activities is the connectivity capacity. In many mining operations, the topographical and climatic conditions are extreme, and connectivity does not exist or is scarce with the usual short-range systems such as Bluetooth and WiFi. For these cases, there are other alternative systems that allow long-range connectivity even in kilometers such as Zigbee, LTE, LoRa, and XBee, among others. This is how, for example, within underground mines, it is possible to apply long-range connectivity technologies together with types of mesh networks to ensure the correct functioning of the sensors under an Internet of Things (IoT) technology.

The implementation of sensors in conjunction with the use of the Internet of Things (IoT) is achieving a very large impact on the mining sector, where it can be very well used for its effective management of mining operations, where it can be applied and not limited to the following cases:Underground mining: Vehicular tracking of haul trucks, scoops, drills, and personnel trucks.Traffic lights inside the mine.Tracking of people at all important control points of the mine.Monitoring of fans and controlling toxic gases—Machine to Machine (M2M).Security control in explosive stores.Control of water levels and automatic control of water pumps.Micro seismic control systems.Vehicle collision avoidance and personnel safety.Control the use of personal protection elements (PPEs).Control of water and energy resources.Mobile control in underground operation.Centralized and distributed control of the operation.Visibility of the operation on the mine front.

The mining operations will have first-rate advanced analytical data capturing data and information from the terrain/field in real time, where an integrated operations center will analyze information, the latter being the brain of the mining operation, where it will also be possible to carry out simulations of operating scenarios under different conditions, through digital twins, thus allowing better proactive and non-reactive decisions to be made, where the decision-making process is carried out much faster to optimize mining processes by making them more efficient and safe [109].

Through the joint implementation of sensors, artificial intelligence (AI), data analytics, Internet of Things (IoT), automation, augmented/virtual reality, robotization, and digital twins, it will be possible to simulate scenarios of mining production processes in real time, which will allow to analyze and study how different parameters or operating variables can modify or alter the performance of the mining operation, allowing this to go continuously from the real world to the meta-verse and return to the real world (physical–virtual–physical), making proactive decisions in favor of improving performance indicators [110].

Undoubtedly, the objectives of the mining operation will be focused on carrying out a safer productive activity focused on the occupational health of its collaborators and promoting sustainability, considering reducing water consumption, using clean renewable energies, eliminating the use of energy and fossils fuels, reducing the amount of waste generated, and protecting their communities and territorial environment [111].

The implementation of technology and innovation for better management of information through sensors, digitization, and automation in mining will ensure that the best proactive decisions are made in any aspect of the mining business, not only based on the economy and greater productivity of its value chain (profits and cost reduction) but also based on safe mining with an emphasis on its human capital (risks and occupational health) and green mining focused on the sustainability of its daily activities (energy, water, and waste).

The scenarios of technological advances will be dizzying in the coming decades, changing many current concepts and paradigms by the year 2023, for which it will be necessary to be professionally prepared and thus understand and be part of the era of Industry 4.0 or perhaps of a future Industry 5.0 paradigm [112].

## 5. Conclusions

The digital transformation revolution reaches all sectors, and among them, it reaches the mining industry. This consists of digitizing the control and operation processes in the value chain using sensors, integrating them into traditional equipment (the existing ones) and capturing data from them, and applying data processing with software. This is already possible today.

Our vision as authors of this article is that in the near future, an intelligent and connected mining operation will be created that will transform huge amounts of data captured by sensors into predictive intelligence, and the result will be a fully integrated, systematized, and definitely machine-learning (ML) operation, which will offer entirely new levels of security, stability, and predictability, considerably reducing the uncertainties that traditionally characterize mining today.

One of the challenges of mining is to maximize production (considering the ratio of metal to ore), reduce capital costs and operating costs, increase the level of safety in daily work, and at the same time, minimize environmental impact. The adoption of intelligent Industry 4.0 technologies will allow mining to improve its productivity by reducing or eliminating various sources of uncertainty and at the same time reducing risks. On the one hand, the planning and coordination of activities are improved, in order to mitigate the uncertainty caused by external forces; on the other hand, the variability of the operation of mining-metallurgical industrial processes is reduced.

For mining countries to take full advantage of the opportunities of smart mining, it is not only necessary to advance in the development of digital infrastructure, technologies, and data management but also challenges associated with the management and coordination of stakeholders of the territory must be resolved, where they carry out their mining operations, promote participatory processes of the community, and thus also become green mining.

The adoption of low-cost sensors in the mining industry offers a safer mine site for workers, predictable mining operations, an interoperable environment for information and communications technology (ICTs) systems and traditional and modern devices, automation for reducing human intervention and enables surveillance in both surface and/or underground mines through operational technology (OT) and information technology (IT).

According to the findings found in this research, it is possible to conclude that the scientific literature does not report on the costs of sensor technologies. In order to carry out benchmarking and also generate competitiveness and massification of these sensor technologies, it is necessary to report the implementation costs in different site conditions. Therefore, it is relevant to mention that this information should be published in the future in scientific articles, for the dissemination of the knowledge of academics, students, the mining industry, and the community.

Significant research challenges and directions are open for future study of mining applications, such as mobility management, scalability, IoT virtualization, digital twins, smart mine enablement through automation, interoperable systems, data distribution, and visibility in real time the status of mining operations.

To protect the health and safety of workers, mining companies have accelerated technologies to reduce the number of people on site and increase the ability to operate mines remotely. Therefore, establishing smart, green, and sustainable mines has gradually become a consensus in the global industry, where recent years have witnessed rapid progress in smart mining. Cutting-edge technologies such as the Internet of Things (IoT), big data, artificial intelligence (AI), machine learning (ML), 5G internet, edge computing, and virtual reality have greatly promoted the intelligent development of the mining industry and comprehensively improved the level sustainability of mining operations; therefore, we can say that the world mining industry is going through a stage of a new revolution.

Finally, in this context, the mining industry faces the challenge of rethinking its production processes with the potential to generate innovations in production systems on an unprecedented scale, increasing their productivity, generating a positive and sustainable impact in the territories where they are located carries out the mining activity, re-imagining the way of doing mining with value chains that promote carbon neutrality and sustainability, supporting with key commodities (Copper, Lithium, and Rare Earth Elements (REEs), among others) for the implementation of clean renewable energy technologies, improving the well-being of people, and taking care of the planet’s ecosystems in a scenario of adaptation and mitigation of climate change.

## Figures and Tables

**Figure 1 sensors-23-06846-f001:**
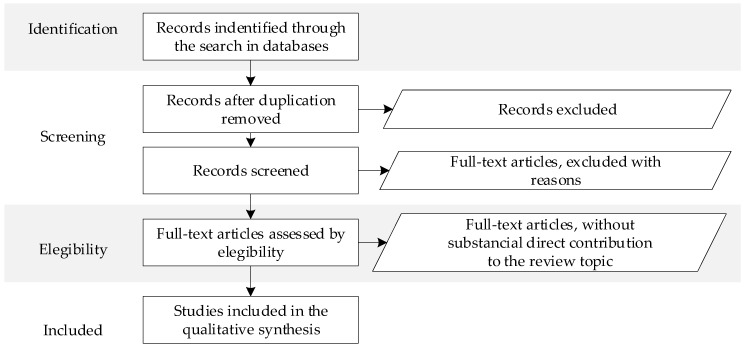
The PRISMA flow diagram of the search process for the highlighted and reviewed articles from WoS and Scopus databases.

**Figure 2 sensors-23-06846-f002:**
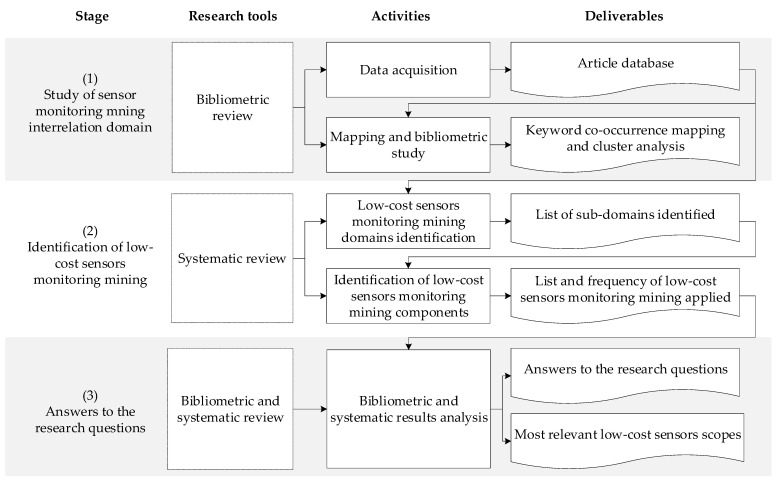
Summary of the methodological procedure implemented in this research. Adapted from [51].

**Figure 3 sensors-23-06846-f003:**
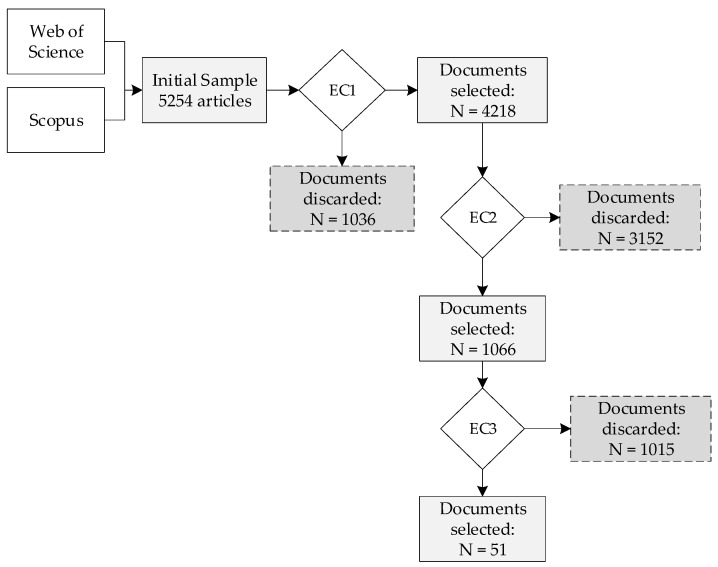
Article selection flowchart.

**Figure 4 sensors-23-06846-f004:**
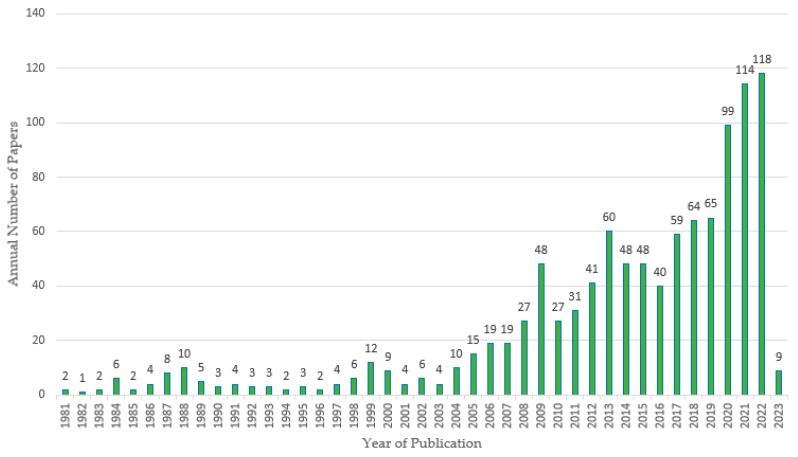
Number of relevant articles published considering 1066 articles selected from 1981 to 2023.

**Figure 5 sensors-23-06846-f005:**
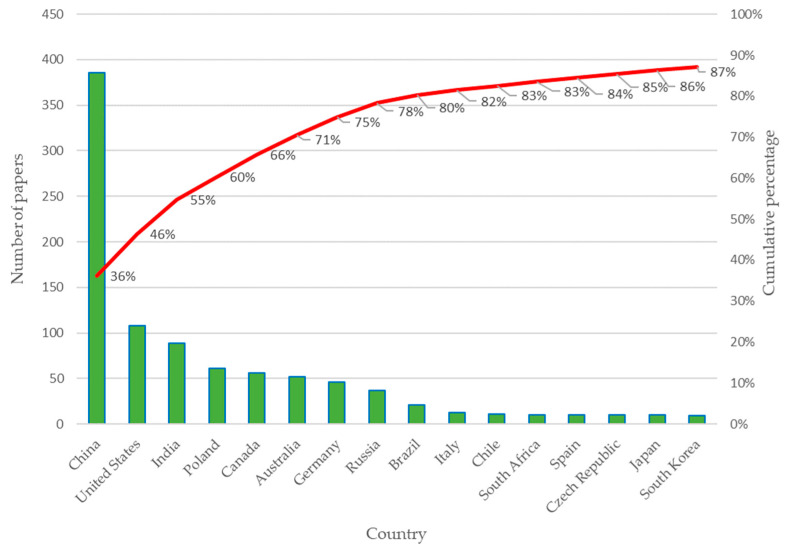
Productivity by country: Number of papers considering 1066 articles selected.

**Figure 6 sensors-23-06846-f006:**
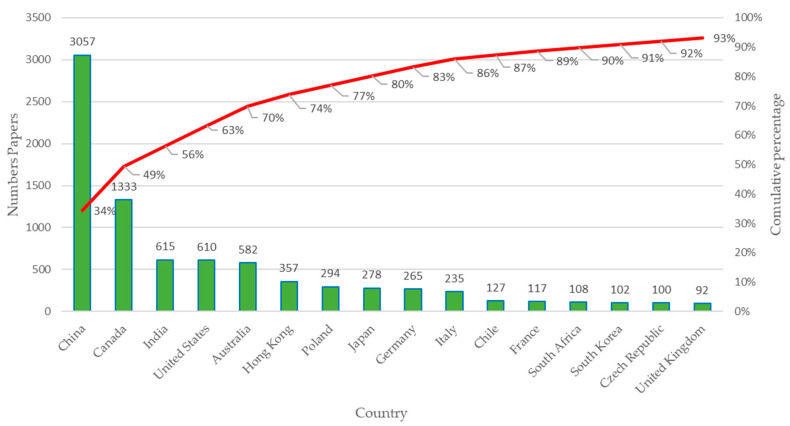
Productivity by country: Number of citations considering 1066 articles selected.

**Figure 7 sensors-23-06846-f007:**
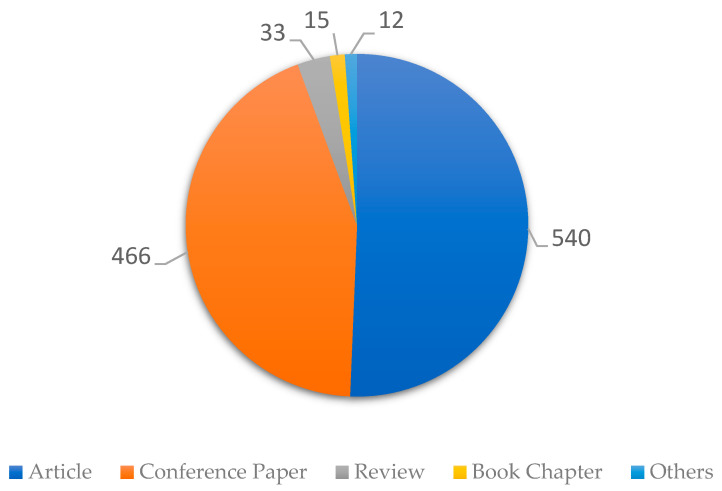
Distribution of documents by type considering 1066 articles selected.

**Figure 8 sensors-23-06846-f008:**
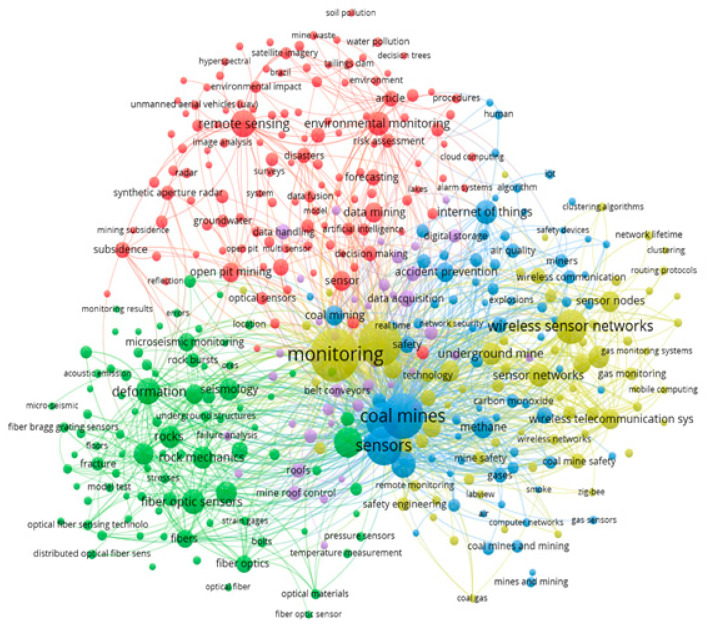
Keyword co-occurrence map using VOSviewer without time dimension, considering 1066 articles selected.

**Figure 9 sensors-23-06846-f009:**
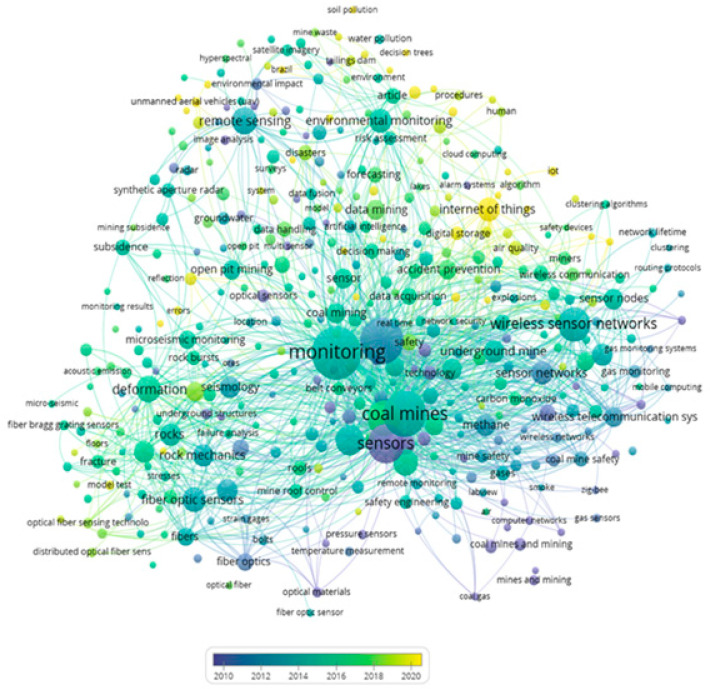
Keyword co-occurrence map using VOSviewer with time dimension, considering 1066 articles selected.

**Figure 10 sensors-23-06846-f010:**
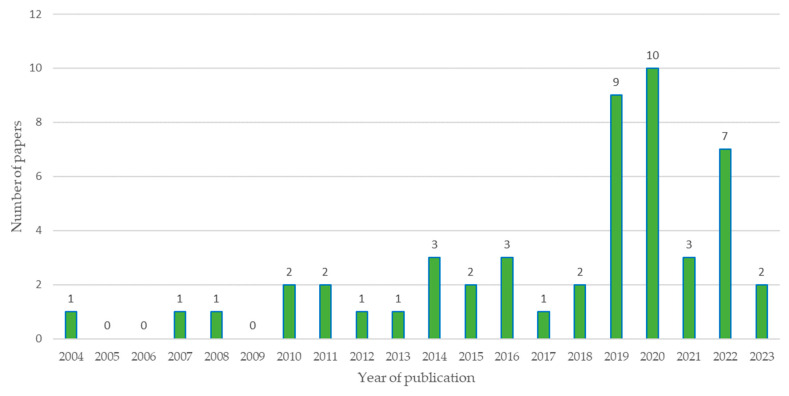
Number of relevant articles published considering 51 articles selected from 2004 to 2023.

**Figure 11 sensors-23-06846-f011:**
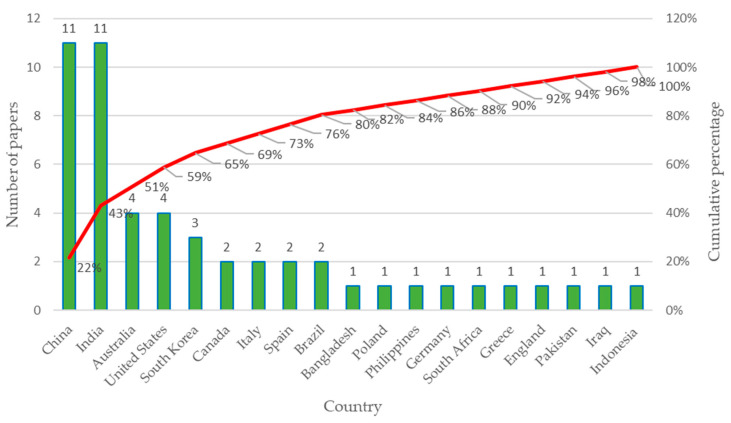
Productivity by country: Number of papers considering 51 articles selected.

**Figure 12 sensors-23-06846-f012:**
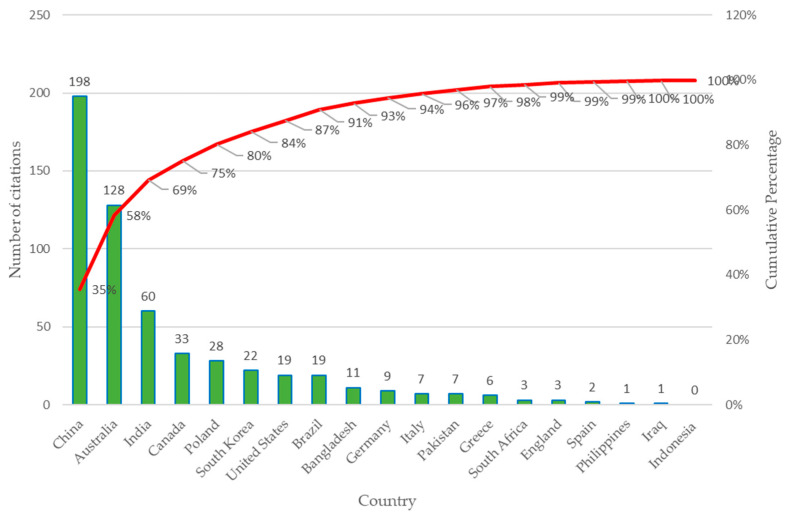
Productivity by country: Number of citations considering 51 articles selected.

**Figure 13 sensors-23-06846-f013:**
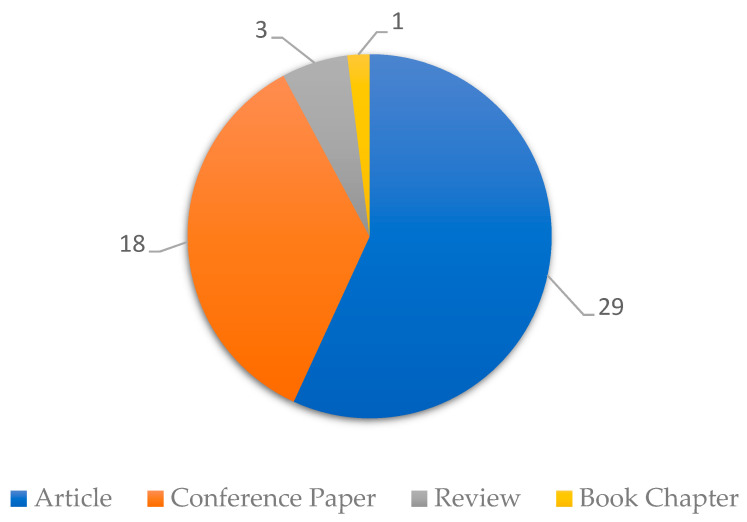
Distribution of documents by type considering 51 articles selected.

**Figure 14 sensors-23-06846-f014:**
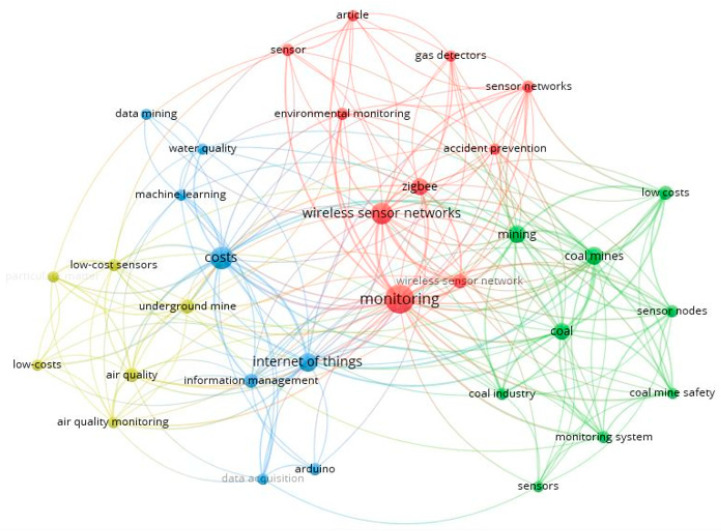
Keyword co-occurrence map using VOSviewer without time dimension, considering 51 articles selected.

**Figure 15 sensors-23-06846-f015:**
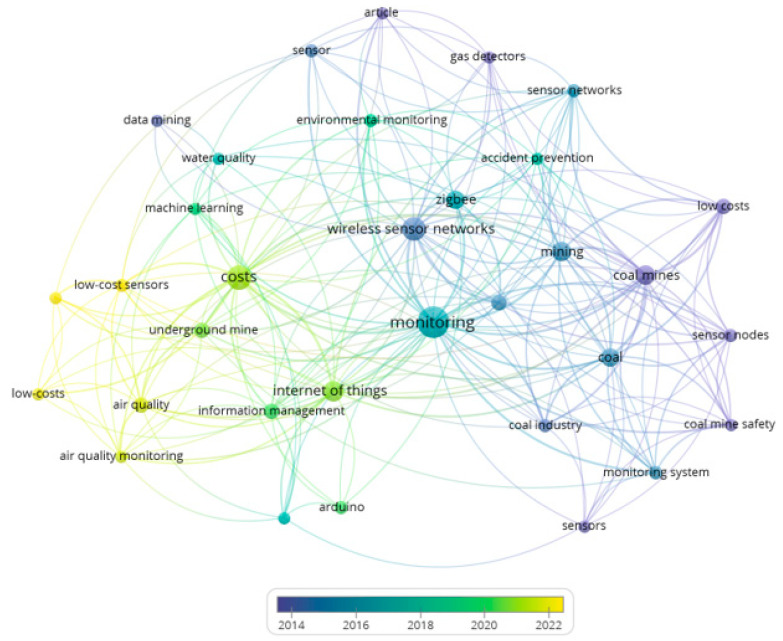
Keyword co-occurrence map using VOSviewer with time dimension, considering 51 articles selected.

**Figure 16 sensors-23-06846-f016:**
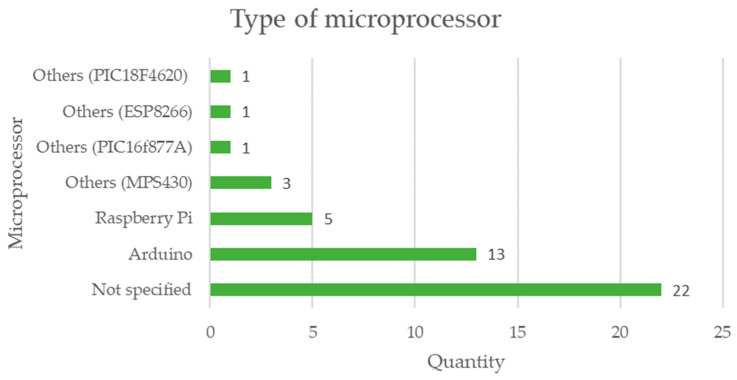
Microcontroller technologies that are implemented in low-cost sensors are applied in mining considering the review of scientific literature.

**Figure 17 sensors-23-06846-f017:**
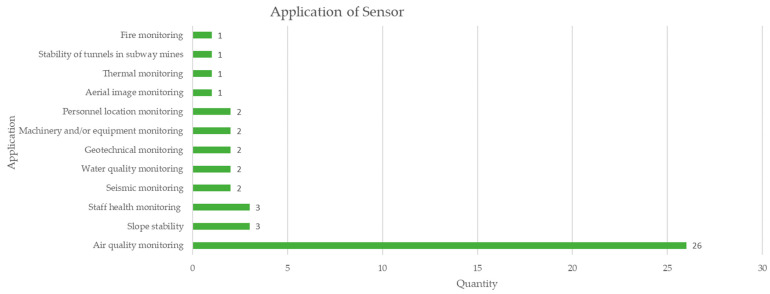
Uses and applications of low-cost sensors applied in mining according to the review of scientific literature.

**Figure 18 sensors-23-06846-f018:**
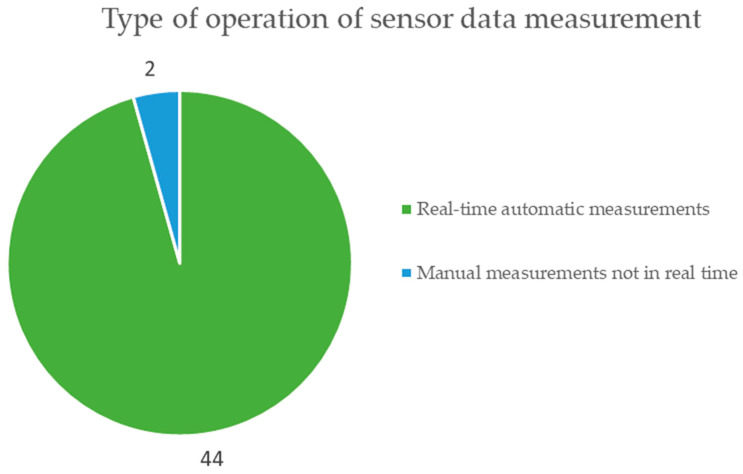
Operation of taking measurements with low-cost sensor technologies for monitoring mining activities according to the review of scientific literature.

**Figure 19 sensors-23-06846-f019:**
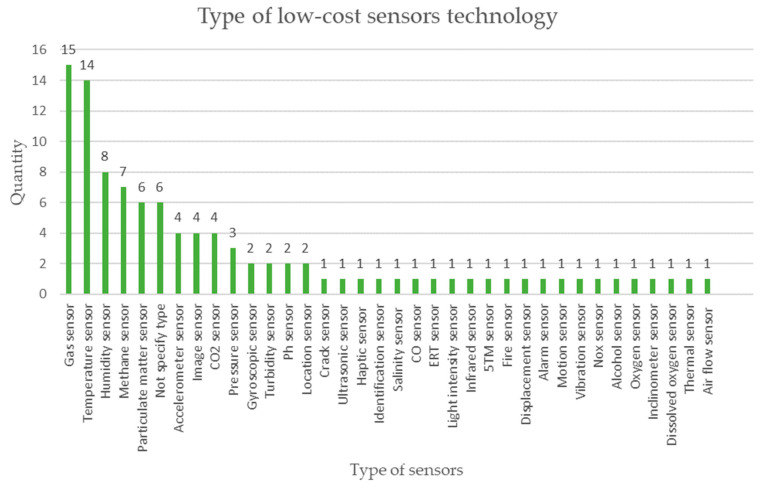
The type of low-cost sensor technology for monitoring mining activities is applied according to the review of scientific literature.

**Figure 20 sensors-23-06846-f020:**
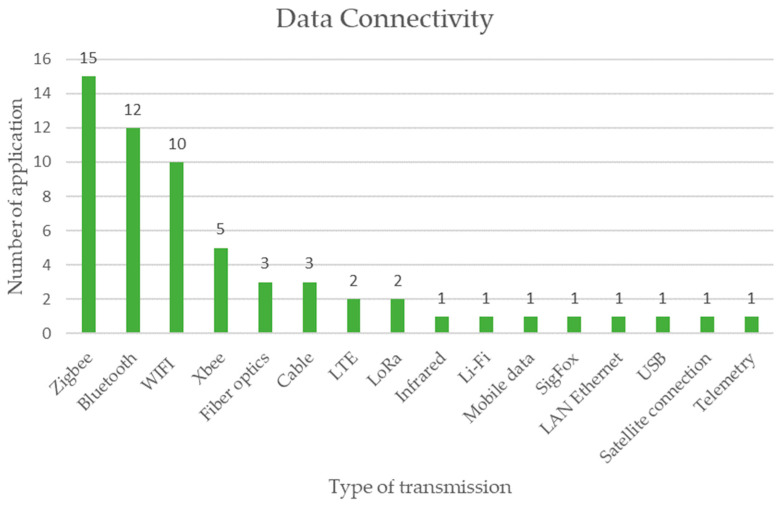
Connectivity of low-cost sensors for monitoring mining activities according to the review of scientific literature.

**Figure 21 sensors-23-06846-f021:**
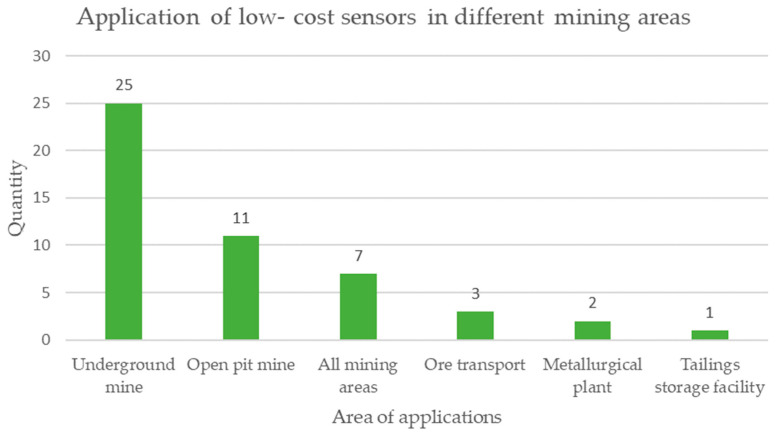
Areas within a mining operation where low-cost sensors are applied for monitoring mining activities according to the review of scientific literature.

**Figure 22 sensors-23-06846-f022:**
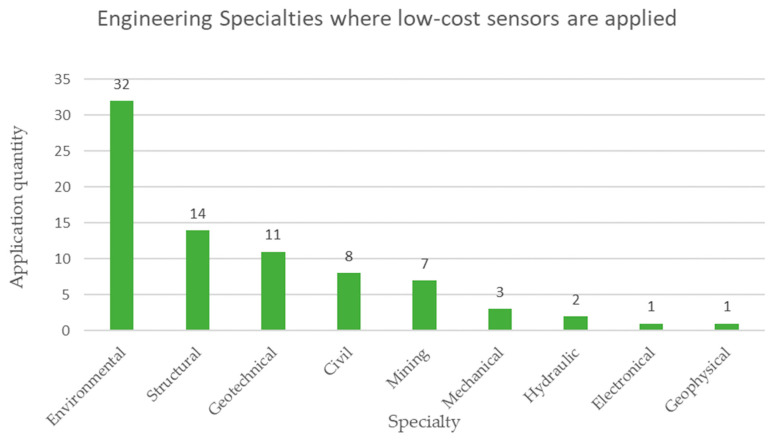
Main engineering specialties where low-cost sensors are applied for monitoring in mining according to the review of scientific literature.

**Figure 23 sensors-23-06846-f023:**
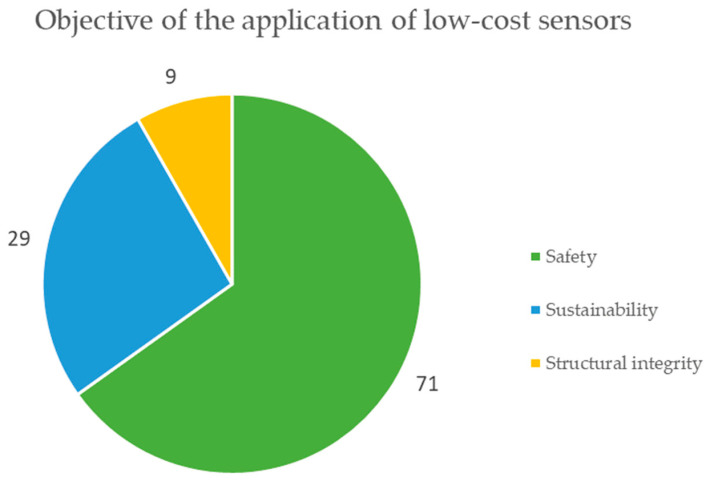
The main application objectives of low-cost sensors implemented in mining according to the review of scientific literature.

**Figure 24 sensors-23-06846-f024:**
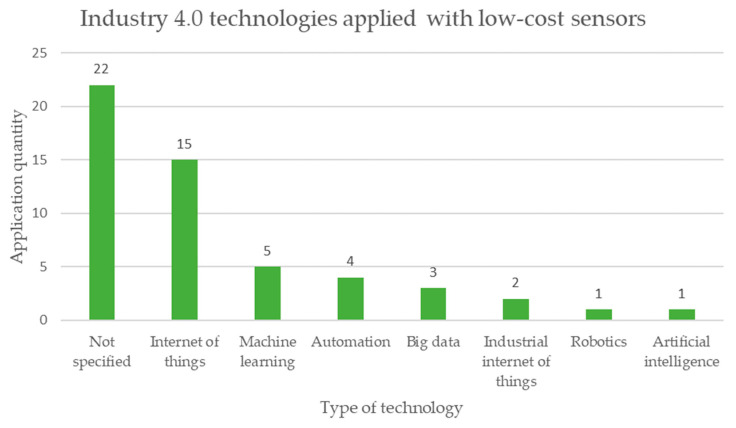
Technologies related to Industry 4.0 are currently being applied in low-cost sensors for monitoring mining activities according to the review of scientific literature.

**Figure 25 sensors-23-06846-f025:**
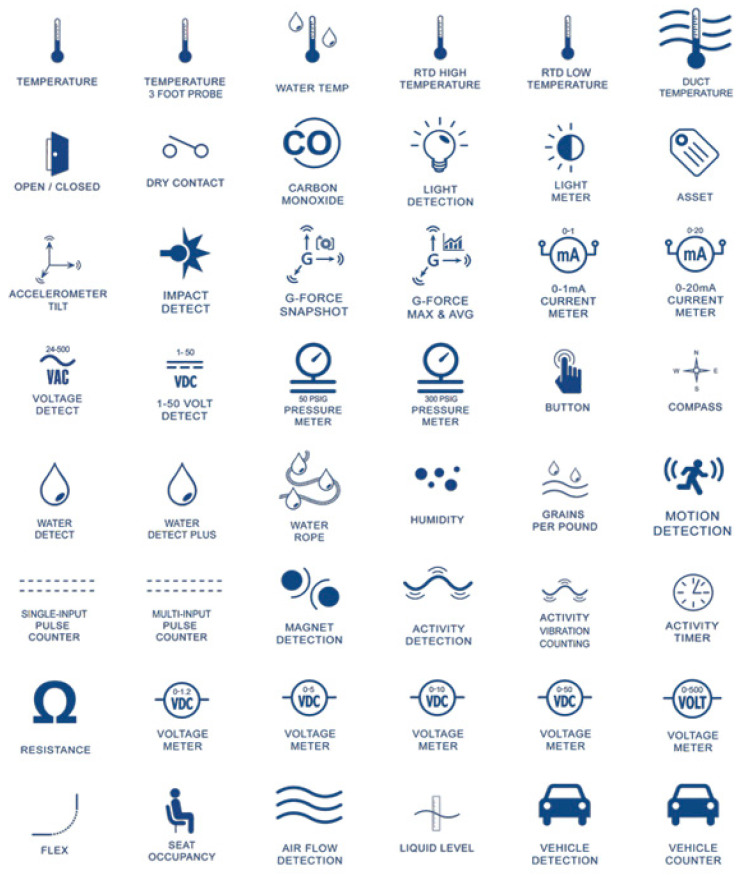
Examples of uses of sensors in mining under an Industry 4.0 paradigm.

**Table 1 sensors-23-06846-t001:** Keywords and Boolean operators used to search relevant articles from Scopus and Web of Science databases.

Keyword/Terms	Boolean Operator	Keyword/Terms	Boolean Operator	Keyword/Terms
Sensor				
Low-cost sensor	AND	Monitoring	AND	Mining
Arduino				
Raspberry Pi				

**Table 2 sensors-23-06846-t002:** Data extraction form considered in this research. Adapted from [51].

ID	Approach	Field	Question	Value
DEM 1	Metadata-perspective	Title	What is the name?	Name
DEM 2	Metadata-perspective	Authors	Who are the authors?	Authors List
DEM 3	Metadata-perspective	Year	Which is the publication year?	Year
DEM 4	Metadata-perspective	Country	Which is the country of the first author?	Country
DEM 5	Metadata-perspective	Journal	Which is the journal?	Journal Name
DEM 6	Metadata-perspective	Document Type	What is the name of the type of document?	Conference paper OR Article OR Review OR Book chapter OR Other
DEM 7	Metadata-perspective	Keywords	Which are the keywords?	Keywords
DEM 8	Metadata-perspective	Citation Count	How many citations have the document?	Number
DEC9	Content-based perspective	Popular Clusters	RQ1: How knowledge about low-cost sensors applied to mining is clustered and how have they evolved over time?	e.g., low-cost sensors mining, among others
DEC10	Content-based perspective	Microcontroller Technology	RQ2: What microcontroller technologies are implemented in low-cost sensors applied in mining?	e.g., Arduino, Raspberry Pi, among others
DEC11	Content-based perspective	Sensor Uses	RQ3: What are the uses of low-cost sensors applied in mining?	e.g., environmental, geotechnical, and geomechanical monitoring, among others
DEC12	Content-based perspective	Operation of Measurements	RQ4: How is the operation of low-cost manual or automatic sensors for monitoring mining activities?	e.g., manual measurements (not in real time), and/or automatic measurements (in real time)
DEC13	Content-based perspective	Sensor Technology	RQ5: What kind of low-cost sensor technology is applied for monitoring mining activities?	e.g., temperature sensor, strain sensor, air quality sensor, among others
DEC14	Content-based perspective	Sensor Connectivity	RQ6: How is the connectivity of low-cost sensors for monitoring mining activities?	e.g., with the use of cables, fiber optics, or wireless such as WiFi, Zigbee, or Bluetooth, among others.
DEC15	Content-based perspective	Mining Area Application	RQ7: What are the areas within the mining operation where low-cost sensors are applied to monitor mining activities?	e.g., mine area (surface or underground), mining waste area, and transportation area, among others
DEC16	Content-based perspective	Engineering Specialties	RQ8: What are the main engineering specialties where low-cost sensors are applied for mining monitoring?	e.g., civil, mechanical, geotechnical, structures, hydraulic, electronic, electrical, mining, metallurgy, environmental, among others
DEC17	Content-based perspective	Sensor Objective	RQ9: What are the main application objectives of low-cost sensors implemented in mining?	e.g., monitoring of sustainability, structural integrity, and/or safety, among others
DEC18	Content-based perspective	Technologies in the Industry 4.0	RQ10: What are the technologies related to Industry 4.0 that are currently being applied in low-cost sensors for monitoring mining activities?	e.g., data mining, big data analytics, machine learning, Internet of Things, artificial intelligence, automation, robotics, and digital twins, among others

**Table 3 sensors-23-06846-t003:** VOSviewer clusters without time scale dimension considering 1066 articles selected.

Cluster Identification	Keywords from Co-Occurrence Analysis	Cluster Interpretation
Cluster “Yellow”	Monitoring, sensor networks, sensor nodes, underground mine	Underground mine monitoring
Cluster “Blue”	Coal mines, sensors, Internet of Things (IoT), methane	Coal mine monitoring
Cluster “Red”	Environmental monitoring, remote sensing, data mining, artificial intelligence	Sustainability monitoring
Cluster “Green”	Deformation, seismology, rock mechanics, fiber optic sensors	Structural monitoring
Cluster “Purple”	Accident prevention, data acquisition, mine roof control	Safety monitoring

**Table 4 sensors-23-06846-t004:** VOSviewer clusters with time scale evolution considering 1066 articles selected.

Cluster Identification	Keywords	Average Year of Publication	Cluster Interpretation
Cluster “Blue”	Sensors, sensor networks, wireless telecommunication, fiber optics, mine safety, gases	2010–2014	Safety monitoring with sensor networks
Cluster “Green”	Coal mines, monitoring, wireless sensor networks, remote sensing, environmental monitoring, fiber optics sensors	2014–2018	Sustainability monitoring with wireless sensor networks and remote sensing
Cluster “Yellow”	Internet of Things (IoT), digital storage, unmanned aerial vehicles (UAV)	2018–2020	Digital mine monitoring with the use of IoT

**Table 5 sensors-23-06846-t005:** VOSviewer clusters without time scale dimension “low-cost sensor—monitoring—mining search”.

Cluster Identification	Keywords	Cluster Interpretation
Cluster “Red”	Monitoring, wireless sensor networks, Zigbee, environmental monitoring	Environmental mine monitoring with wireless low-cost sensors
Cluster “Blue”	Costs, Internet of Things (IoT), Arduino, machine learning (ML), data mining	Low-cost sensor monitoring with Industry 4.0 technologies
Cluster “Green”	Mining, coal mine safety, monitoring system, sensor nodes, low-cost	Safety monitoring in coal mines with low-cost sensors
Cluster “Yellow”	Underground mine, air quality monitoring, low-cost sensors	Miners’ health monitoring in underground mines with low-cost sensors

**Table 6 sensors-23-06846-t006:** VOSviewer clusters with time scale evolution “low-cost sensor—monitoring—mining search”.

Cluster Identification	Keywords	Average Year of Publication	Cluster Interpretation
Cluster “Blue”	Wireless sensor networks, sensors, coal mine safety, monitoring system, low-costs	2010–2014	Coal mine safety monitoring with wireless low-cost sensor networks
Cluster “Green”	Monitoring, Zigbee, environmental monitoring, water quality, accident prevention	2014–2018	Environmental monitoring with Zigbee technology
Cluster “Yellow”	Internet of Things (IoT), Arduino, air quality monitoring, low-cost sensors	2018–2020	Use of low-cost sensors to monitor air quality with Internet of Things (IoT) technology

**Table 7 sensors-23-06846-t007:** Comparison of the main characteristics of the articles selected dealing with low-cost sensors for monitoring mining activities. The following abbreviations are considered for the Microprocessor (Raspberry Pi, R, Arduino, A, Other, O, or Not Specified, NS), Operation (Real Time Automatic, RTA or Not Real Time Manual, NRTM), Connectivity (Zigbee, Z, Bluetooth, B, WiFi, W or Other, O), Mining Area (Underground mine, UM, Open pit mine, OPM, All areas, AA, Ore Transport, OT, Metallurgical Plant, MP, or Tailings Storage Facility, TSF), Engineering specialty (Environmental, E, Structural, S, Geotechnical, G, Civil, C, Mining, M, Mechanical, ME, Hydraulic, H, or Electronical, E), Application Objective (Safety, S, Sustainability, SU, or Structural Integrity, SI), Industry 4.0 technology (Internet of Things, IoT, Machine learning, ML, Automation, A, Big data, BD, Industrial Internet of Things, IIoT, Robotics, R, Artificial Intelligence, AI, or Not specified, NS), and Cost (USD).

Reference	Microprocessor	Operation	Connectivity	Mining Area	Engineering Specialty	Application Objective	Industry 4.0	Cost
Xiao et al. [83]	NS	RTA	O	UM	E	S	NS	-
Zafra-Perez et al. [84]	NS	RTA	O	OPM	E	S	IoT	-
Zietek et al. [85]	A	RTA	Z	UM	E	SU	IoT	-
Mellors et al. [86]	NS	RTA	W	UM	E	S	NS	-
Amoah et al. [87]	A	RTA	Z	UM	E	S	IoT	-
Kim et al. [88]	A	RTA	Z, B, W	AA	E, S, G	S, SU, SI	IoT	-
Amoah et al. [89]	A, R	RTA	Z	UM	E	S	IoT	-
Wang [90]	NS	RTA	Z	UM	E	S	NS	-
Fu et al. [91]	NS	RTA	Z	UM	E	S, SU	ML	-
Ali et al. [92]	R	RTA	Z	UM	E	S, SU	IoT, IIoT	-
Zhou et al. [93]	A	RTA	Z, W	UM	E	S, SU	IoT, IIoT	-
Sharma and Maity [94]	A	RTA	W	UM	E	S	IoT	-
Aguirre-Jodré et al. [95]	NS	RTA	Z, W	OPM	M	S, SI	IoT	-
Mardonova and Choi [96]	A	RTA	Z	UM	E	S, SI	IoT	-
Bui et al. [97]	NS	NRTM	W	OPM	M	SI	NS	-
Chehri and Saadne [98]	NS	RTA	Z, W	UM	E	S	NS	-

## Data Availability

The data presented in this study are available on request from the corresponding author.

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
