# Peer review of "Low-Cost Sensors Technologies for Monitoring Sustainability and Safety Issues in Mining Activities: Advances, Gaps, and Future Directions in the Digitalization for Smart Mining"

_sensors, 2023, doi:10.3390/s23156846_

Round 1
Reviewer 1 Report
Overall an important contribution to literature around industry 4.0. The paper would benefit from more historical background or context around mining in 1.0, 2.0 and 3.0 transitions. Provide some lit review on that and a discussion comparing the unique features of industry 4.0 compared to other transitions.
Next, i think you can't talk about technology without a consideration of how the mining industry is fragmented by small, medium and large mines. Each have different capabilities. Small and medium mines has historically been out of the market for many mining technology tools. Does industry 4.0 provide them more opportunities?
Some figures are poor quality and need better resolution.
Why not include digitization as a key word for your search?

A few minor grammar issues.
Reviewer 2 Report
The topic of the manuscript is interesting given the relevance that low-cost hardware (including sensors) and software has acquired in different scientific and industrial fields; in addition, this topic fits the scope of the Journal. After a careful revision, the following comments are provided for the enhancement of the manuscript.
The list of keywords is limited to ten terms, according to the template of the Journal.
For a wider perspective about the applicability of Arduino and Raspberry Pi, it is suggested to mention recent publications which deal with usage of these devices together with sensors out of mining scenario. This could be conducted in the subsection 1.2 and some papers are provided for consideration by the authors if they agree with the suggestion:
- Artificial vision wireless PV system to efficiently track the MPP under partial shading. International Journal of Electrical Power & Energy Systems 2023. https://doi.org/10.1016/j.ijepes.2023.109198
- Smart Farming Revolution: Portable and Real-Time Soil Nitrogen and Phosphorus Monitoring for Sustainable Agriculture. Sensors 2023, 23, 5914. https://doi.org/10.3390/s23135914
- IoT real time system for monitoring lithium-ion battery long-term operation in microgrids. Journal of Energy Storage, 2022. https://doi.org/10.1016/j.est.2022.104596
- Development of a Raspberry Pi-Based Sensor System for Automated In-Field Monitoring to Support Crop Breeding Programs. Inventions 2021, 6, 42. https://doi.org/10.3390/inventions6020042
As the authors know, a common practice in scientific papers consists on placing a final paragraph in the Introduction indicating the structure of the rest of the manuscript. This helps the reader to find the information in a clear manner. This issue must be revised in the paper.
The second section describes very well the followed search and selection procedure. In particular, tables 1 and 2 are very illustrative.
When a figure is cited in the main body of the manuscript, it must start capitalized, for example, in line 255, it should be “In Figure 2…”.
Regarding the achieved results, in the figure 7 (pie chart), it is suggested to enlarge the number for a better readability. The same comment is applicable to figures 13, 18 and 23.
The paragraph after table 4 is well written and portrays the evolution of sensors and sensor networks, at least, from the humble perspective of this reviewer.
In the subsection 3.5.2, Raspberry Pi is considered as a microprocessor. This is not rigorously true given the fact that such a device is a single board computer, based on microprocessor. In the same sense, Arduino is an electronics prototyping platform based on microcontroller.
In figure 20, LAN appears. This reviewer supposes that it is referred to a Ethernet or TCP/IP-based LAN. If possible, the authors should try to specify this aspect. Indeed, this abbreviation should be included in the Abbreviations list.
The format of references must be slightly revised according to the template of the Journal.
Round 2
Reviewer 2 Report
The new version of the paper has properly addressed the reviewer concerns.